# Development of Dl1.72, a Novel Anti-DLL1 Antibody with Anti-Tumor Efficacy against Estrogen Receptor-Positive Breast Cancer

**DOI:** 10.3390/cancers13164074

**Published:** 2021-08-13

**Authors:** Gabriela Silva, Joana Sales-Dias, Diogo Casal, Sara Alves, Giacomo Domenici, Clara Barreto, Carolina Matos, Ana R. Lemos, Ana T. Matias, Khrystyna Kucheryava, Andreia Ferreira, Maria Raquel Moita, Sofia Braga, Catarina Brito, M. Guadalupe Cabral, Cristina Casalou, Duarte C. Barral, Pedro M. F. Sousa, Paula A. Videira, Tiago M. Bandeiras, Ana Barbas

**Affiliations:** 1iBET, Instituto de Biologia Experimental e Tecnológica, Apartado 12, 2781-901 Oeiras, Portugal; jsdias@ibet.pt (J.S.-D.); giacomo.domenici@ibet.pt (G.D.); clara.barreto@medicina.ulisboa.pt (C.B.); carolina.matos@ibet.pt (C.M.); ar.lemos@ibet.pt (A.R.L.); kkucheryava@ibet.pt (K.K.); mraquelmoita@ibet.pt (M.R.M.); anabrito@ibet.pt (C.B.); pedrosousa@ibet.pt (P.M.F.S.); tiagob@ibet.pt (T.M.B.); ana.barbas@bayer.com (A.B.); 2Instituto de Tecnologia Química e Biológica António Xavier, Universidade Nova de Lisboa, Av. da República, 2780-157 Oeiras, Portugal; 3Departamento de Anatomia, NOVA Medical School (NMS), Universidade NOVA de Lisboa, 1150-082 Lisbon, Portugal; diogo.casal@nms.unl.pt (D.C.); sara.p.alves@chlc.min-saude.pt (S.A.); 4iNOVA4Health, CEDOC, NOVA Medical School (NMS), Universidade NOVA de Lisboa, 1150-082 Lisbon, Portugal; ana.matias@nms.unl.pt (A.T.M.); andreia.ferreira@nms.unl.pt (A.F.); sofia.braga@jmellosaude.pt (S.B.); guadalupe.cabral@nms.unl.pt (M.G.C.); cristina.casalou@ucd.ie (C.C.); duarte.barral@nms.unl.pt (D.C.B.); 5Serviço de Anatomia Patológica, Centro Hospitalar de Lisboa Central-Hospital de São José, 1150-199 Lisbon, Portugal; 6Unidade de Mama, Instituto CUF de Oncologia, 1998-018 Lisbon, Portugal; 7UCIBIO, Departamento Ciências da Vida, Faculdade de Ciências e Tecnologia, Universidade NOVA de Lisboa, 2829-516 Caparica, Portugal; p.videira@fct.unl.pt

**Keywords:** ER^+^ breast cancer, Notch signaling, DLL1, monoclonal antibody, cell proliferation, angiogenesis, tumor growth

## Abstract

**Simple Summary:**

Over 70% of breast cancers (BCs) are estrogen receptor-positive (ER^+^). The development of endocrine therapy has considerably improved patient outcomes. However, there is a clinical need for novel effective therapies against ER^+^ BCs, since many of these do not respond to standard therapy, and more than one-third of responders acquire resistance, experience relapse and metastasize. The Notch ligand Delta-like 1 (DLL1) is a key player in ER^+^ BC development and aggressiveness. Contrary to complete Notch pharmacological inhibitors, antibody-targeting of individual Notch components is expected to have superior therapeutic efficacy and be better tolerated. In this study, we developed and characterized a novel specific anti-DLL1 antibody with efficacy in inhibiting BC cell proliferation, mammosphere formation and angiogenesis, as well as anti-tumor and anti-metastatic efficacy in an ER^+^ BC mouse model without side effects. Thus, our data suggest that this anti-DLL1 antibody is a promising candidate for ER^+^ BC treatment.

**Abstract:**

The Notch-signaling ligand DLL1 has emerged as an important player and promising therapeutic target in breast cancer (BC). DLL1-induced Notch activation promotes tumor cell proliferation, survival, migration, angiogenesis and BC stem cell maintenance. In BC, DLL1 overexpression is associated with poor prognosis, particularly in estrogen receptor-positive (ER^+^) subtypes. Directed therapy in early and advanced BC has dramatically changed the natural course of ER^+^ BC; however, relapse is a major clinical issue, and new therapeutic strategies are needed. Here, we report the development and characterization of a novel monoclonal antibody specific to DLL1. Using phage display technology, we selected an anti-DLL1 antibody fragment, which was converted into a full human IgG1 (Dl1.72). The Dl1.72 antibody exhibited DLL1 specificity and affinity in the low nanomolar range and significantly impaired DLL1-Notch signaling and expression of Notch target genes in ER^+^ BC cells. Functionally, in vitro treatment with Dl1.72 reduced MCF-7 cell proliferation, migration, mammosphere formation and endothelial tube formation. In vivo, Dl1.72 significantly inhibited tumor growth, reducing both tumor cell proliferation and liver metastases in a xenograft mouse model, without apparent toxicity. These findings suggest that anti-DLL1 Dl1.72 could be an attractive agent against ER^+^ BC, warranting further preclinical investigation.

## 1. Introduction

Breast cancer (BC) is the most frequent cancer in women and the second leading cause of cancer deaths in women worldwide [1]. BC is a highly heterogeneous and multifactorial disease that can be classified into four subtypes (Luminal A/B, human epidermal growth factor receptor 2 (HER2)-positive and triple-negative), based on the expression of the estrogen receptor (ER), progesterone receptor (PR) and HER2 [1,2]. Clinical outcomes differ greatly between subtypes, and most BC deaths arise from distant metastasis in the liver, bone and lung [3]. ER^+^ BC accounts for more than 70% of all BC cases. The development of endocrine therapies such as tamoxifen and fulvestrant has considerably improved the outcomes of ER^+^ BC patients [4]. Nonetheless, approximately one-third of patients do not respond to initial therapy, and a large proportion of tumors acquire resistance, recur, and undergo metastasis [2,5,6]. Therefore, there is an urgent need to develop new therapeutic strategies to improve the prognosis of patients with ER^+^ BC.

In this regard, the Notch signaling pathway appears as an attractive alternative therapeutic target in BC. This pathway is a cell–cell communication system that is crucial for embryonic development and post-natal tissue homeostasis and regeneration [7]. Mammalian Notch signaling relies on four receptors (Notch1–4) and five activating ligands: Delta-like (DLL) 1, 3 and 4 and Jagged (JAG) 1 and 2 [8,9]. Notch ligands are transmembrane proteins with an extracellular domain (ECD) that contains an N-terminal domain, a highly conserved Delta/Serrate/LAG-2 region and several epidermal growth factor (EGF)-like repeats [8,9]. Binding of the ECD of one of the ligands to one of the receptors present in an adjacent cell triggers sequential proteolytic cleavages that release the Notch intracellular domain (NICD) in the cytoplasm. The NICD then translocates to the nucleus, where it interacts with DNA-binding proteins, leading to the transcription of Notch target genes (e.g., *HEY* family), which regulate many cellular processes [10]. The activation of Notch signaling varies remarkably depending on pathway component expression, signal dose and cell context [11]. The dysregulation of Notch signaling, by virtue of abnormal expression of its receptors and/or ligands, has been reported in more than 50% of BCs [12], and aberrant Notch signaling is implicated in essentially all hallmarks of cancer [13] and drug resistance [6,14,15].

The human Notch ligand DLL1 is overexpressed in BC tumors and is associated with poor prognosis in patients with ER^+^ BCs [16,17]. We and others have shown that DLL1 contributes to BC tumor biology through the promotion of cancer cell colony formation, cell proliferation, survival, migration, and invasion, BC stem cell (BCSC) function, metastases formation and angiogenesis [17,18,19]. Moreover, recently, DLL1^+^ cells have been shown to bear similarities to CSCs by showing a high tumor-initiating capacity as well as the ability to drive metastasis and chemoresistance in aggressive luminal breast tumors [20]. These multiple roles of DLL1 in BC support the development of specific anti-DLL1 therapies for ER^+^ BC treatment. DLL1-targeting therapies are expected to provide clinical benefits when used alone and in combination with conventional chemotherapy, or as an alternative therapeutic strategy in the case of endocrine resistance.

In this study, we characterized the in vitro and in vivo BC anti-tumorigenic effect of a novel anti-DLL1 antibody, the Dl1.72, developed using phage display technology. Biophysical characterization of Dl1.72 showed that it binds to human DLL1 with nanomolar affinity, displaying no binding for other human Notch ligands. Cellular assays using MCF-7 ER^+^ BC cells revealed that Dl1.72 impaired DLL1-Notch signaling and reduced cancer cell proliferation and migration as well as the BCSC population. In addition, endothelial cell tube formation ability was impaired by Dl1.72, suggesting a potential anti-angiogenic effect of this mAb. Finally, using a human ER^+^ BC xenograft model, we verified that Dl1.72 remarkably inhibited tumor growth, reducing both tumor cell proliferation and liver metastasis, without associated toxicity. Together, these observations suggest that anti-DLL1 Dl1.72 could be an attractive tool against ER^+^ BC and warrant further preclinical investigation of this mAb against ER^+^ BC.

## 2. Materials and Methods

### 2.1. Cell Culture and Reagents

HEK293E6 [21] and MCF-7 (ATCC, Manassas, VA, USA, HTB-22) cells were cultured as previously described [18]. CHO-K1 cells (ATCC, CCL-61) were cultured in DMEM F12 media (#12634010) containing 10% heat-inactivated fetal bovine serum (FBS, #10270-106) plus 100 µg/mL penicillin and streptomycin (#15140163) (all from Gibco Thermo Fisher Scientific, Waltham, MA, USA). HUVEC cells (C-003-5C) were cultured in Medium 200 (M-200-500) plus large vessel endothelial supplement (A14608-01) (all from Life Technologies, Waltham, MA, USA) and used until passage four. All cells were cultured at 37 °C, 5% CO_2_, according to the supplier’s instructions, and tested for the absence of mycoplasma. Culture media was replaced every 2–3 days. The Notch pathway signaling inhibitor DAPT (Sigma-Aldrich, St. Louis, MO, USA, D5942) was used at 5 μM. DMSO (Sigma-Aldrich, #472301) was used as a control for DAPT in vitro exposure in a final dilution more than 1/1000.

### 2.2. Phage Display Screening, Antibody Generation and Characterization

Selection of anti-DLL1 single-chain variable fragments (scFvs) by phage display and generation of anti-DLL1 Dl1.72 were performed as described previously [22].

For Dl1.72 production, HEK293E6 suspension cells were transfected with 0.5 mg/L of the respective LC and HC plasmids. An isotype-matched negative control mAb (Ctr Ab) was also produced in a similar mode [22]. The secreted mAbs were purified in endotoxin-free conditions and analyzed by sodium dodecyl sulfate–polyacrylamide gel electrophoresis (SDS–PAGE), analytical size exclusion chromatography and Western blotting, as described in [22]. Protein concentration was determined by the Bradford method and spectrophotometry at 280 nm. Purity was estimated from SDS–PAGE by densitometry analysis using ImageJ/Fiji software [23], and endotoxin content was measured using the Endosafe-PTS system (Charles River, Boston, MA, USA, PTS2001F). Dl1.72 reactivity and specificity were evaluated by ELISA (see Appendix A for proteins used) and flow cytometry, as described in [22]. Binding kinetic characterization of D11.72 interaction with untagged rhDLL1, containing the full ECD [22], was performed by Surface Plasmon Resonance (SPR) Horseradish Peroxidase conjugation of anti-DLL1 Dl1.72, and Ctr Ab was performed using the EZ-Link™ Plus Activated Peroxidase kit (Thermo Scientific, Waltham, MA, USA, #31488); the resulting conjugates were analyzed by SDS–PAGE, Western blotting and ELISA. For details, see Appendix A.

### 2.3. Notch Reporter Assay

MCF-7 cells (1.25 × 10^5^ cells/mL) were co-transfected with 750 ng of Notch firefly luciferase reporter (4xCSL, Addgene, Watertown, MA, USA, #41726) along with 10 ng of pRL-SV40 vector encoding renilla luciferase (Promega, Madison, WI, USA, #E2231) using GeneJuice transfection reagent (Merck-Millipore, Darmstadt, Germany, #70967), according to the manufacturer’s instructions. Cells were then plated in 96-well plates either not coated (control) or pre-coated with rhDLL1- Fc or Fc control protein in the absence or presence of either Dl1.72, Ctr Ab (1, 2.5, 5, 10 and 20 µg/mL each), DAPT or an equivalent amount of vehicle (DMSO. Luciferase activities were measured in whole cell lysates 28 h after culture using the Dual Luciferase assay kit (Promega, #E1910), following the kit’s instructions, in a plate reader MultiskanTM FC (Thermo Scientific). Firefly luciferase activities were normalized to renilla luciferase units in the same samples. Three to four replicates were analyzed for each condition. Results were calculated relative to cells plated in the control uncoated wells, which were set as 1.

### 2.4. Real-Time RT-qPCR

Cells were plated on culture plates either not coated (control) or pre-coated with rhDLL1-Fc, for the induction of DLL1-Notch target genes, or Fc protein in the absence or presence of Dl1.72, Ctr Ab, DAPT, or DMSO, as indicated elsewhere. Total RNA was extracted using the RNeasy Mini kit (Qiagen, La Jolla, CA, USA, #50974104), and cDNA was generated from equal amounts of RNA by reverse transcription using the Advantage RT-for-PCR kit (Clontech Laboratories, Mountain View, CA, USA, # 639506), as per the manufacturer’s instructions. The expression of *HEY-1*, *HEY-L*, *HPRT1*, *RPL22*, *SOX2* and *SOX9* genes was quantified on Roche LightCycler 480 equipment using SYBR Green I Master Kit (Roche, Bazel, Switzerland). mRNA transcripts were normalized to housekeeping genes *HPRT1* and *RPL22* levels in the same sample, and the results were calculated as fold change relative to control cells, as previously described [24]. The primers used in these assays are listed in Appendix A.

### 2.5. MCF-7 Cell Growth, Scratch Wound Healing, and Mammosphere Formation Assays

Cells were seeded at 3–4 × 10^4^ cells/cm^2^ in 24-well plates (500 µL/well) or 96-well plates (100 µL/well) either not coated or pre-coated with rhDLL1-Fc or Fc proteins, as indicated above. Cells were left untreated or were treated with either Dl1.72, Ctr Ab (both at 10 μg/mL), DAPT, or DMSO and incubated for 70–76 h. After incubation, cell growth was determined using the MTT (3-(4,5-dimethylthiazolyl-2)-2,5-diphenyltetrazolium bromide) colorimetric method (96-well plates) or by microscopy using the trypan blue exclusion method (24-well plates), as previously described [18,25]. Each condition was tested in triplicate (trypan blue exclusion) or quadruplicate (MTT) for all assays. Growth media was replaced with fresh media containing the specific treatment every 48 h.

For the wound healing assay, MCF-7 cells (2.5–4 × 10^4^ cells/cm^2^) seeded in 24-well plates were treated 16 h later with Abs or left untreated as above and cultured for 48–52 h until 90% confluence. Thereafter, cell monolayers were scratched, washed and incubated in complete media with the respective treatments. Images of 3 different areas of the wound of each scratch were taken at 0 and 72 h after the scratches were made with a Leica DMI6000 microscope (5× objective). The wound widths were determined using ImageJ/Fiji software, as described in [18].

The mammosphere (MS) formation assay was performed as indicated in [22].

### 2.6. HUVEC Tube Formation, Cell Growth, and Viability Assays

Tube formation assay was performed using the Angiogenesis starter kit from Life Technologies (#A1460901) following the manufacturer’s instructions, with Geltrex from the kit or Matrigel (BD Biosciences, San Jose, CA, USA, #356230). Briefly, HUVEC at 80% confluence were collected using trypsin/EDTA and trypsin neutralizing solution (Lonza, Basel, Switzerland, CC-5012 and CC-5002, respectively) and seeded at 4 × 10^3^ cells/cm^2^ in T-25 tissue culture flasks in medium 200 with large vessel endothelial supplement. After 6 h, cells were left untreated (control) or were treated with either Dl1.72, Ctr Ab (10 µg/mL each), DAPT or DMSO and incubated for 3 days until 80% confluence. Afterwards, cells were collected and seeded on a layer of solidified Geltrex or Matrigel at a density of 2.5 · 10^4^ cells/cm^2^ in complete media (250 µL/well of 24-well tissue culture plates) and incubated at 37 °C for 16–18 h. Pictures of cells stained with Calcein AM (Corning, NY, USA, #354216, 2 µg/mL in PBS plus Ca^++^ and Mg^++^, 250 µL/well, 30 min at 37 °C) were taken using a phase-contrast inverted microscope (5× objective). At least four independent images were acquired per condition. Morphology of the tube network formation, including the number of nodes and meshes, and total branching length were quantified using the ImageJ Angiogenesis Analyzer plugin [26]. The growth and viability of cells at the end of the treatments were evaluated by microscopy using the trypan blue exclusion method, as described above.

### 2.7. Human Breast Tumor Samples

Formalin-fixed, paraffin-embedded (FFPE) ER^+^ BC 2.5 µm tissue sections (*N* = 9) were obtained from surgical BC specimens, selected by pathologists from the Hospital Professor Doutor Fernando Fonseca (HFF, Amadora-Sintra, Portugal). The study was approved by the ethics committees from NOVA Medical School, Faculdade de Ciências Médicas, Universidade NOVA de Lisboa, and HFF (ref. 98/2019/CEFCM). All participants gave informed consent, and this study was performed in accordance with the ethical standards outlined in the 1964 Declaration of Helsinki. The clinicopathological characteristics of BC samples used in the study are described in Appendix A.

### 2.8. MCF-7 Breast Cancer Xenograft Model

All animal experiments were conducted with approval from the national regulatory authority, Direção Geral de Alimentação e Veterinária (DGAV) (ref. 0421/000/000/2017) and the Institutional Ethics Committee (CEFCM 07/2016) and performed according to their guidelines and under the rules of the Federation for Animal Science Associations (FELASA). Five-week-old female NOD scid gamma (NSG) female mice (Charles River Laboratories) were used for this study and maintained under pathogen-free conditions. Four days prior to tumor cell implantation, estrogen pellets (0.36 mg/pellet 90 day release, Innovative Research of America, 0.36 mg/pellet 90-day release, Innovative Research of America, Sarasota, FL, USA, #NE-121) were implanted subcutaneously (s.c.) in the mice dorsal neck region using a sterilized trocar. Xenografts were initiated by s.c. implanting 2 × 10^6^ MCF-7 cells in 100 µL of PBS:matrigel (1:1 ratio, BD Biosciences, #356230) into the mammary fat pad of each tested mouse (2 mammary fat pad injections/mouse) using an established protocol [27]. After fifteen days, mice were weighed and injected intraperitoneally (i.p.) with Dl1.72, Ctr Ab (each at 10 mg/kg of mice body weight (BW), *n* = 5 mice per each group) or an equal volume of vehicle PBS (*n* = 6 mice). Treatment was repeated twice a week for 12 weeks. Animals were monitored daily for clinical symptoms and adverse effects. Tumor growth and mouse weight were determined weekly. Tumor volume was estimated by measuring tumor length (L) and width (W) with a digital caliper and by applying the following formula: (π × L × W2)/6. Twelve weeks after the beginning of treatment, mice were sacrificed and their blood, tumor, liver and kidney were collected for analysis.

### 2.9. Serology

Mice blood withdrawal was performed by cardiac puncture to tubes. To stop coagulation, 10 units of heparin per mL of blood were added to collection tubes. Blood was centrifuged at 2000× *g* for 10 min at 4 °C, and serum was immediately collected. Urea, creatinine and alanine aminotransferase (ALT) serum levels were determined by DNAtech (Lisbon, http://www.dnatech.pt/web/, accessed on 18 October 2019).

### 2.10. Histological and Immunohistochemical Analysis

Immunohistochemical (IHC) staining of FFPE human ER^+^ BC tissue section specimens was performed as previously specified [28]. Stained samples were scored for both signal intensity and abundance of positive cells using ImageJ/Fiji software using a macro developed by the authors (https://github.com/ClaraBarreto/Dl1.72.git, accessed on 22 July 2021). Staining intensity was measured using the scale 0–3, with 0 being negative and 3 being very high expression. The H-score was calculated by multiplying intensity with abundance using the following formula: (1 × (% cells 1+) + 2 × (% cells 2+) + 3 × (% cells 3+)). Scoring was done independently by three researchers.

Mice tumor, liver, and kidney tissues were fixed in formalin (10%), embedded in paraffin and cut into 4 µm sections on a microtome (Leica, Wetzlar, Germany). The sections were stained with hematoxylin-eosin (H&E) [29] or immunostained with the pan-cytokeratin AE1/AE3/PCK26 antibody cocktail (#760-2135) for the identification of human epithelial cells or anti-human Ki67 (#05298512001) for detection of proliferating cells. IHC was performed by BenchMark ULTRA—Automated IHC and in situ hybridization slide staining systems (Ventana, Roche), as described in [30]. Stained images were scanned using NanoZoomer—SQ Digital Slide Scanner (Hamamatsu, Japan). The tumor area was determined from the largest tumor sections stained with H&E and AE1/AE3/PCK26 Ab and metastasis of BC cell in liver sections stained with AE1/AE3/PCK26 Ab using the imageJ/Fiji software. The percentage of proliferating cells from tumor sections stained with Ki67 Ab was determined by ImageJ/Fiji software with the ImmunoRatio plugin [31]. Proliferation and metastasis analysis were done by examining multiple random fields of view (FOVs) for each sample. Histological analyses was independently evaluated by two pathologists, and immunostaining analysis by at least two researchers. For details on IHC staining, see Appendix A.

### 2.11. Analysis of Tumor Cell Apoptosis by TUNEL

Mice tumor sections prepared as above were deparaffinized with Xylene (3 min, 3 times) and rehydrated in 100% ethanol (3 min, 2 times), 95% ethanol (3 min), 70% ethanol (3 min) and ddH_2_O (10 min). For antigen retrieval, samples were soaked in Citrate Buffer, pH 6, heated in a microwave until liquid boiled, and cooled down for 20 min. Then, samples were washed with Tris-buffered saline containing 0.05% Tween-20 (5 min, 3 times) and incubated with proteinase K (Merck Millipore, # 21627, 20 µg/mL, 15 min, RT). Then, the ApopTag red in situ apoptosis detection kit (Merck Millipore, # S7165) procedure for fluorescent staining of paraffin-embedded tissue samples was followed according to the manufacturer’s instructions. Samples were mounted in ProLong Gold antifade reagent with DAPI for nuclei staining (Invitrogen, Waltham, MA, USA, #P36931). Images were acquired on a Leica DMI 6000-inverted microscope equipped with a Leica DFC360 FX camera, using a 20x HCX PL FLUOTAR objective, controlled with the Leica Application Suite × software. Images were analyzed with the ImageJ/Fiji software. The percentage of apoptotic cells was determined by counting 10 independent FOVs, performed by independent researchers.

### 2.12. Statistical Analysis

Results are presented as mean ± standard deviation (SD) or mean ± standard error of the mean (SEM) for the mice studies (see Figure 1e). Statistical analyses were performed using Graphpad Prism 8 software. The significance of differences was analyzed using the Student *t*-test and the non-parametric Mann–Whitney test for normally and non-normally distributed datasets, respectively. One-way analysis of variance (ANOVA) with Tukey’s test was used to determine differences between tumor growth datasets. *p* values < 0.05 were considered statistically significant.

## 3. Results

### 3.1. Characterization of the Generated Anti-DLL1 Dl1.72 Antibody That Specifically Interacts with DLL1

Using Phage display technology to generate synthetic mAbs with blocking activity against DLL1 [22], we identified our scFv lead candidate, which was converted into the anti-DLL1 Dl1.72 full IgG1 antibody. Dl1.72 was produced in mammalian cells in parallel with a control isotype IgG1 mAb (Ctr Ab). The purified mAbs have the expected sizes of approximately 50 kDa for IgG HC and 25 kDa for IgG LC, with a purity > 90% and less than 0.16 EU/mg. To evaluate Dl1.72 binding ability for its cognate antigen, we performed a dose-dependent ELISA of Dl1.72 or Ctr Ab with a fixed concentration of rhDLL1-Fc. Cross-reactivity to murine DLL1 (rmDLL1 presents more than 85% sequence identity to hDLL1) was also evaluated. As a negative control, Dl1.72 was incubated with Fc protein. Dl1.72 bound to both purified human and murine DLL1 proteins but not to the Fc control protein, with no binding of Ctr Ab (Figure 1a). Specificity testing by ELISA in the same conditions showed Dl1.72 does not bind to any other human Notch ligand (rhDLL3, rhDLL4, rhJAG1 and rhJAG2) (Figure 1a). The calculated half-maximal effective concentration (EC_50_) of Dl1.72 to purified rhDLL1 was found to be 1.15 ± 0.16 nM (Table 1). The binding kinetics of Dl1.72 interaction with rhDLL1 were measured by surface plasmon resonance (Figure 1b). The kinetic affinity (K*_D_*) of the interaction between D11.72 and rhDLL1 was calculated to be 4.78 ± 1.05 nM (Table 1).

To verify the binding ability of Dl1.72 to cell surface hDLL1, CHO-K1 cells overexpressing hDLL1 and control cells without hDLL1 [22] were incubated with various concentrations of Dl1.72 or Ctr Ab, and binding was analyzed by flow cytometry. Dl1.72 bound strongly to DLL1-overexpressing cells but not to cells not expressing DLL1. Ctr Ab showed no effective binding to both cell lines (Figure 1c). The EC_50_ of the interaction between Dl1.72 and cells overexpressing hDLL1 was calculated to be 20.7 pM, on average (Table 1). This EC_50_ value is more than 50-fold lower than the one obtained with the immobilized pure protein by ELISA, suggesting that the native protein conformation in the cells as well as the amino acid residues recognized by Dl1.72 are more accessible in the cell surface DLL1.

We next examined the ability of Dl1.72 to recognize and bind to endogenous human DLL1 by immunohistochemistry using cross-sections of breast tissues from patients with ER^+^ BC. Matched BC tissue samples were incubated with Dl1.72, Ctr Ab or a commercially available rabbit anti-DLL1 antibody (ab84620), used as a control for DLL1 detection and to confirm tissue distribution, as previously reported [17]. Mammary tissues, especially BC tissues, contain significant amounts of human IgGs [32]. Thus, Dl1.72 and Ctr Ab were previously conjugated with HRP for these assays. Our results showed high binding levels of Dl1.72 to ER^+^ BC tissues, contrary to Ctr Ab. Moreover, Dl1.72 and ab84620 presented similar IHC staining patterns to the same tissue samples, suggesting that both Abs are binding to the same antigen, i.e., DLL1 (Figure 1d and Appendix A). Quantification of the intensity and abundance of each Ab staining showed that Dl1.72 presented a higher immunoreactivity score in comparison with ab84620 (Figure 1e). Altogether, these results suggest that anti-DLL1 Dl1.72 binds specifically to endogenous DLL1 in BC tissue samples.

### 3.2. Anti-DLL1 Dl1.72 Inhibits DLL1-Notch Signaling in ER^+^ Breast Cancer Cells

DLL1-Notch signaling has been implicated in the oncogenic capacities of MCF-7 ER^+^ BC cells [18]. Accordingly, this BC cell line was chosen to examine Dl1.72 capacity to impair DLL1-Notch signaling by luciferase Notch reporter assay and evaluating the expression levels of the Notch target genes *HEY-1* and *HEY-L,* which are significantly induced by DLL1 [18]. Dl1.72-treated cells showed significant dose-dependent inhibition of DLL1-Notch signaling when compared to control cells not treated or treated with Ctr Ab (Figure 2a). Similar to DAPT, Dl1.72 at 10–20 µg/mL almost completely inhibited Notch signaling activation by rhDLL1. As expected, Fc control protein had no effect in the Notch reporter activity (Figure 2a).

Consistent with the impairment of Notch reporter activity, Dl1.72 decreased the induction levels of *HEY-1* and *HEY-L* by DLL1 in a dose-dependent manner (Appendix A). At 10 µg/mL, this mAb reduced the levels of DLL1-induced *HEY-1* and *HEY-L* by 20% and 60%, respectively (Figure 2b) when compared to control cells not treated or treated with Ctr Ab. Furthermore, DAPT reduced *HEY-1* and *HEY-L* levels by 55% and 80%, respectively, in response to the ligand (Figure 2b). Assessment of the effect of Dl1.72 in the mRNA levels of these genes in unstimulated MCF-7 cells showed Dl1.72 at 10 µg/mL decreased the expression of basal *HEY-L* at levels comparable with DAPT (30% reduction on average when compared to control cells) but had no significant effect on those of *HEY-1* (Figure 2c). Altogether, these results show that anti-DLL1 Dl1.72 is able to impair DLL1-Notch signaling.

### 3.3. Anti-DLL1 Dl1.72 Decreases MCF-7 ER^+^ Breast Cancer Cell Proliferation, Migration and Mammosphere Formation

We have shown that siRNA-mediated DLL1 downregulation decreases MCF-7 cell proliferation and migration, whereas incubation with DLL1 increases them [18]. Following the findings described above, showing that Dl1.72 impairs DLL1-Notch signaling in MCF-7 cells, we next evaluated the Dl1.72 ability to impair DLL1-induced MCF-7 cell proliferation, using the MTT assay and trypan blue exclusion methods. In agreement with our previous results [18], MCF-7 cells exposed to DLL1 alone or in the presence of Ctr Ab or DMSO showed a significant increase in cell growth (45% and 100% increase by MTT and trypan blue exclusion, respectively) compared to control cells or cells treated with Fc control protein. In contrast, treatment with Dl1.72 mAb reduced MCF-7 cell growth induced by DLL1 by approximately 50% (Figure 3a,b). DLL1-induced cell growth was partially inhibited by DAPT (Figure 3a,b). Dl1.72 also delayed the proliferation of unstimulated cells by 25% on average compared to control cells, as determined by trypan blue exclusion by microscopy (Figure 3c). The reduction was comparable with that obtained with DAPT. Trypan blue exclusion showed similar low levels of non-viable cells in all tested conditions, suggesting that Dl1.72 indeed impaired cell proliferation.

The ability of Dl1.72 to impair MCF-7 migratory capability was assessed in scratch wound-healing assays. Dl1.72 resulted in a significant delay in wound closure in MCF-7 cells (20% reduction on average) compared to control cells not treated or treated with Ctr Ab (Figure 3d,e), suggesting that this mAb impairs MCF-7 cell migration.

Breast cancer stem cells (BCSCs), which have tumor initiation potential, have emerged as drivers of tumor chemoresistance, recurrence and metastasis in BC [33,34,35]. Given that DLL1-Notch signaling in ER^+^ BCs promotes BCSC function [17], we next examined the potential of Dl1.72 to influence the BCSC subpopulation of MCF-7 cells. To assess this, we employed the mammosphere (MS) formation assay, a functional 3D-BCSC cell culture-based assay commonly used to evaluate in vitro the amount of BCSC within a heterogeneous cell population [36]. The amount of MS formed correlates with the content of BCSCs. The results obtained in these assays showed that Dl1.72 impaired the number of the MS formed by approximately 30% and 38% compared to control cells not treated or treated with Ctr Ab, respectively (Figure 3f,g). The effect was comparable to that obtained with DAPT. To confirm that MS resulted from the expansion of the BCSC subpopulation, we evaluated the expression levels of the stemness-related genes *SOX2* and *SOX9*, involved in the maintenance and expansion of BCSCs [34,35], in the formed MS and in the original MCF-7 cells growing in parallel as 2D monolayers. As compared to the parental cells cultured in 2D, MS presented three- and nine-fold increased levels of *SOX2* and *SOX9*, respectively (Appendix A), similar to what was reported previously [34,35].

Next, we investigated whether DLL1 has a role in *SOX2* or *SOX9* expression. MCF-7 cells plated in rhDLL1-Fc coated plates showed a clear increase in *SOX9* (five-fold induction on average) but not *SOX2*, when compared to control cells not treated or treated with Fc control protein (Appendix A). We further confirmed these results by challenging DLL1-exposed MCF-7 cells to Dl1.72 and DAPT. In line with an impairment of DLL1-Notch signaling (Figure 2), Dl1.72 reduced the expression levels of *SOX9* induction by DLL1 by approximately 55% when compared to untreated control cells or cells exposed to the same concentration of Ctr Ab (Appendix A). Inhibition of Notch signaling with DAPT nearly eliminated DLL1-induced *SOX9* expression (Appendix A). To our knowledge, this is the first report showing that the DLL1 Notch mediates *SOX9* expression in BC cells.

Overall, these results indicate that anti-DLL1 Dl1.72 can impair oncogenic features of ER^+^ BC MCF-7 cells, such as cell proliferation, migration and the maintenance of the BCSC subpopulation, possibly through *SOX9* modulation.

### 3.4. Anti-DLL1 Dl1.72 Impairs the Angiogenic Potential of Human Vascular Endothelial Cells

Previous studies have shown that DLL1 plays a critical role in tumor angiogenesis [17,37]. We therefore investigated whether Dl1.72 exerts anti-angiogenic effects. For this, we used the HUVEC tube formation assay, a rapid and quantitative in vitro method that measures the ability of vascular endothelial cells to form capillary-like structures on an extracellular matrix substrate widely used as a screening assay for angiogenic or anti-angiogenic agents [38]. Our results showed that Dl1.72 treatment impaired the pro-angiogenic potential of HUVEC cells, in comparison to control cells not treated or treated with Ctr Ab, as evidenced by a significant reduction in the number of capillary-like structures (total branching length and meshes/cavities) as well as the number of nodes (Figure 4a–d). DAPT severely affected the tube network performance of HUVEC cells (Figure 4a–d). To further confirm these findings, we aimed to exclude the possibility that the in vitro anti-angiogenic effects were caused by a Dl1.72-induced reduction in HUVEC cell proliferation and viability. Assessment of cell proliferation and viability in these assays showed that Dl1.72 treatment had no effect on these biological functions of HUVEC cells (Figure 4e,f). Moreover, analysis of the mRNA levels of *HEY-1* and *HEY-L* in HUVEC cells showed that Dl1.72 significantly reduced the levels of *HEY-L* (50% reduction, on average), compared to the untreated control cells or cells treated with Ctr Ab, but not the levels of *HEY-1*. DAPT reduced *HEY-L* and *HEY-1* levels by approximately 60% and 30%, respectively (Figure 4g). Of note, *HEY-L* has been shown to promote tumor angiogenesis [39].

Altogether, these results show that anti-DLL1 Dl1.72 impairs Notch signaling in HUVEC cells and suggest it has antiangiogenic activity, without affecting HUVEC viability.

### 3.5. Anti-DLL1 Dl1.72 Inhibits Tumor Growth in a MCF-7 ER^+^ Breast Cancer Xenograft Mice Model

We next investigated in vivo the ability of Dl1.72 to inhibit tumor growth of human ER^+^ BC in a MCF-7 xenograft mice model, to address the tumorigenic role of DLL1 in human BCs [17] (Figure 5a). Treatment with Dl1.72 significantly impaired tumor growth (Figure 5b–d). Analysis of the largest tumor cross-sections stained with the pan-cytokeratin AE1/AE3 antibody, an epithelial cell marker that stains MCF-7 tumor cells, confirmed a significant decrease in tumor sizes in mice treated with Dl1.72 (Figure 5e,f). Histological analysis showed that tumors from control mice treated with PBS or Ctr Ab presented more necrotic areas within the tumor when compared to mice treated with Dl1.72 and were more prone to invade adjacent tissues, such as muscle and fat, than tumors treated with Dl1.72 (Figure 5g, upper panels). Conversely, there was more fibro-fatty infiltration of the tumor regions in mice treated with Dl1.72 (Figure 5g, lower panels). No body weight loss (Appendix A) or signs of animal distress were observed. Organ function analysis using measurement of the levels of ALT, urea and creatinine in the sera of these mice showed similar levels of these parameters among all groups (Appendix A). Consistent with these results, histological analysis of liver and kidney sections from mice treated with Dl1.72, PBS or Ctr Ab showed no signs of tissue damage (Appendix A). The inhibition of tumor growth of MCF-7 xenografts by Dl1.72 and its safety profile were validated in a second in vivo assay (Appendix A). Collectively, these results show that anti-DLL1 Dl1.72 inhibits tumor growth and is well tolerated.

Since DLL1 promotes tumor cell proliferation [17], and Dl1.72 decreases MCF-7 cell proliferation in vitro (Figure 3a–c), we assessed tumor cell proliferation in tumor sections stained with Ki67 antibody. Our results showed that tumors from mice treated with Dl1.72 presented a significant decrease of proliferating tumor cells (approximately 60% reduction of Ki67^+^ cells) in comparison to tumors from control mice treated with PBS or Ctr Ab (Figure 6a,b).

The observation that DLL1 downregulation induces apoptosis of MCF-7 cells [18] prompted us to next evaluate apoptosis in tumor sections, using the TUNEL method. We verified that 12 weeks after the initiation of the treatments, tumors from mice treated with Dl1.72 showed a significant increase in the percentage of apoptotic cells compared to tumors from control mice that received PBS or Ctr Ab (Figure 6c,d), even though these events were not very frequent (mean 5% apoptotic cells/field in Dl1.72 tumors from treated mice).

Histological analysis of H&E-stained livers suggested the presence of tumor cell micrometastasis in control mice treated with PBS and Ctr Ab but not in mice treated with Dl1.72 mAb. To determine if this was the case, the presence of liver metastases was then evaluated by IHC in mouse liver sections stained with AE1/AE3 antibody. These results showed that Dl1.72 treatment significantly reduced liver metastasis, as determined by the number of metastatic foci and area of metastasis, when compared to livers from mice treated with PBS or Ctr Ab (Figure 6e,f and Appendix A), suggesting Dl1.72 has the capacity to reduce metastasis of ER^+^ BCs.

Collectively, our results show that anti-DLL1 Dl1.72 mAb is effective in inhibiting ER^+^ tumor growth and metastases.

## 4. Discussion

Notch signaling is a well-established oncogenic player in BC [13], and its ligand DLL1 has been shown to be overexpressed in BC tumors. Moreover, recent studies have demonstrated that it is involved in tumor progression, metastasis, chemoresistance, and poor patient survival in ER^+^ BCs [16,17,18,20,40]. Given the relevance of Notch signaling components in multiple aspects of BC biology, various strategies have been pursued to develop therapeutic agents to block the Notch signaling pathway in BC [41]. Gamma-secretase inhibitors (GSIs), small compounds that prevent the release of the NICD [10], were the first and most extensively explored inhibitors of the Notch pathway. While extensive clinical testing showed GSIs have anti-tumor efficacy, these induce severe gastrointestinal tract toxicity in patients, likely due to complete Notch signaling inhibition, limiting their therapeutic benefits [41]. These results demonstrated the need for a specific targeted approach, for example by using anti-Notch signaling component Abs. Over the last decade, Abs have become an important component in the arsenal of cancer therapeutics [42]. Antibodies targeting individual Notch receptors and ligands (i.e., JAG1, DLL4) have been developed, showing anti-tumor efficacy against different types of cancers in preclinical studies, with manageable toxicity profiles [41,43,44,45,46,47,48]. The ER^+^ BC is the most common BC subtype and is treated with anti-estrogenic drugs such as tamoxifen and fulvestrant, presenting a good prognosis. However, a large proportion of ER^+^ BCs develop endocrine resistance, leading to treatment failure, tumor recurrence and eventually metastasis, which dramatically decreases life expectancy [5]. Motivated by these observations and the accumulating evidence suggesting DLL1 drives ER^+^ BC pathogenesis and aggressiveness, we developed specific DLL1 targeting mAbs, which could show activity against this type of BC.

Here we report the development and characterization of Dl1.72, a novel anti-DLL1 mAb selected by phage display. This mAb binds to DLL1 protein with a calculated kinetic affinity of 4.78 nM, but not to the other human ligands of the Notch pathway, and recognizes cellular DLL1 in DLL1-overexpressing cells and ER^+^ BC tumor tissues. Dl1.72 is able to inhibit Notch signaling in MCF-7 ER^+^ BC and HUVEC cells. Moreover, in vitro studies using MCF-7 cells showed that Dl1.72 reduced their cell proliferation and wound closure capacities. These results are in line with our previous data showing that DLL1 promotes MCF-7 proliferation and migration [18]. It is likely that the wound healing impairment by Dl1.72 is due mostly to reduced cell proliferation, given the low migration capacity of these cells [18,49]. Importantly, treatment with Dl1.72 reduced MCF-7 MS forming ability, suggesting that it may be able to reduce the BCSC subpopulation of MCF-7 cells. This finding is in accordance with the enrichment of DLL1 in BCSCs [20] and the reduction of BCSC number and function in ER^+^ BC cells upon the loss of DLL1 [17] and Notch signaling blockade [46]. This is particularly relevant since BCSCs, which have tumor initiation potential, have emerged as drivers of tumor chemoresistance, recurrence and metastasis in BC [33,40]. Interestingly, we also found that Dl1.72 reduced the expression of *SOX9*, driven by DLL1. *SOX9* expression was induced in the MCF-7 MS in our assays, compared to cells growing in 2D, and it was reported to be involved in BC stemness induction as well as playing various pro-oncogenic roles in BC, including in the ER^+^ subtype [34,50]. Based on these observations, it is tempting to speculate that Dl1.72 anti-BCSC activity in MCF-7 cells involves the regulation of *SOX9*. However, this needs to be properly assessed in future studies. Data showing that DLL1^+^ BC tumor cells display CSC characteristics [20], together with our findings showing that DLL1 controls the expression of *SOX9*, supports the importance of these studies, which may provide additional insights into the mechanisms underlying DLL1’s key oncogenic role in BC. *SOX9* is a direct target of Notch signaling [50] and is regulated by DLL1 in other cell types such as pancreatic cells [51], but to our knowledge no previous information is available on the specific role of DLL1 in *SOX9* expression in the context of BC.

DLL1 was previously shown to be involved in BC angiogenesis [17]. As such, we explored the possibility that Dl1.72 mAb could inhibit angiogenesis. Using HUVEC cells as a model, we found that inhibition of the Notch-dependent gene *HEY-L* by Dl1.72 mAb was associated with the reduction of their angiogenic potential. Strikingly, the reduction in angiogenesis potential in vitro by Dl1.72 mAb was not accompanied by an increased cell cytotoxicity, which clearly points to a potential anti-angiogenic effect and is not related to the reduction in cell viability/increased HUVEC cell cytotoxicity. These results are consistent with a role of DLL1-Notch signaling in the promotion of angiogenesis [17,37]. Considering the reported angiogenic role of *HEY-L* in BC [39], our results may suggest a potential link between *HEY-L* downregulation by Dl1.72 and its anti-angiogenic potential, which should be assessed by future studies.

In vivo studies demonstrated that Dl1.72 mAb severely impaired tumor growth of MCF-7 xenografts without detectable associated side effects. Tumor growth inhibition by Dl1.72 was significantly associated with reduction of tumor cell proliferation and apoptosis. These results are consistent with studies showing that DLL1 downregulation impairs tumor growth and tumor cell proliferation in xenografts [17] and MCF-7 proliferation and apoptosis [18], suggesting that decreased proliferation and induction of apoptosis accounted, at least partially, for the Dl1.72 anti-tumor effects. Moreover, our in vitro data showing that Dl1.72 has anti-BCSC activity and anti-angiogenic potential support the hypothesis that these effects of Dl1.72 can also contribute to its anti-tumor activity. It is important to point out that Dl1.72 recognizes murine DLL1, and that DLL1 is important for endothelial cell branching and sprouting [37]. Accordingly, it is conceivable that targeting murine DLL1, namely in the tumor microenvironment, could also account for the anti-tumorigenic effects herein observed in the MCF-7 xenograft mouse model. Importantly, in addition to tumor growth inhibition, we found that Dl1.72 mAb decreased the formation of liver metastases. Over 90% of deaths of BC patients are due to metastases, and liver metastases can potentially occur in aggressive ER^+^ BCs, which are associated with decreased BC patient survival [5]. Considering the role of DLL1 cells in the regulation of BCSCs in ER^+^ tumors [17] and recent findings showing DLL1^+^ tumor cells confer chemoresistance to conventional therapy [20], our findings strengthen the interest in evaluating the anti-cancer potential of Dl1.72 mAb in aggressive metastatic models of ER^+^ BC tumors to further evaluate its therapeutic potential, both as monotherapy or in combination with standard endocrine therapy.

## 5. Conclusions

In conclusion, the present study describes the development and characterization of a novel anti-DLL1 mAb with potent anti-tumor activity in an ER^+^ BC mouse model, without detectable side effects. Given the growing body of evidence showing the crucial role of DLL1 in the aggressive phenotypes of this BC type and its association with high mortality in BC patients, Dl1.72 may prove to be a promising tool for clinical development of a treatment for patients with aggressive ER^+^ BCs, as well as for drug-resistant ER^+^ BCs.

## Figures and Tables

**Figure 1 cancers-13-04074-f001:**
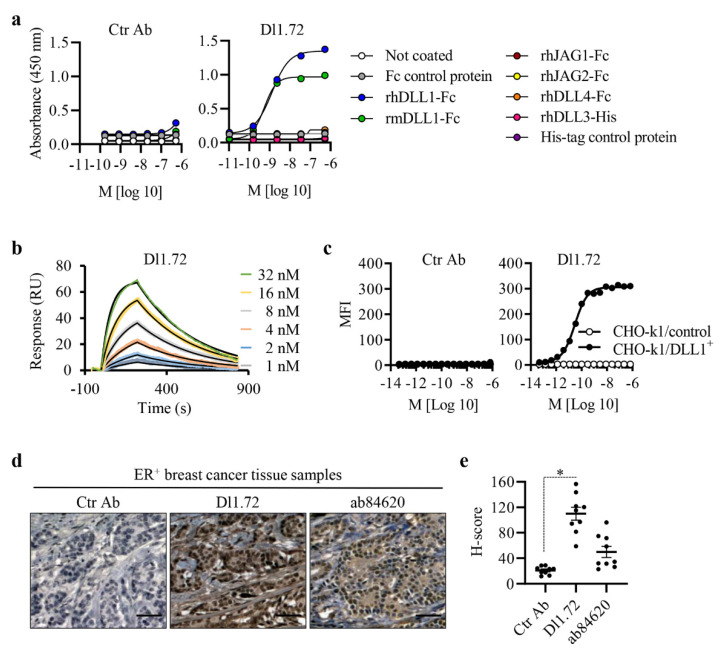
Anti-DLL1 Dl1.72 binding affinity and specificity to DLL1. (**a**) ELISA measuring Dl1.72 and isotope-matched negative control (Ctr Ab) binding to recombinant human (rh) Notch ligands rhDLL1-Fc, rhDLL3-His, rhDLL4-Fc, rhJAG1-Fc and rhJAG2-Fc, and murine DLL1 (rmDLL1). Uncoated wells and wells coated with Fc protein or an unrelated His-tag protein were used as negative controls. (**b**) Surface plasmon resonance was used to determine the Dl1.72 binding affinity to rhDLL1 (without Fc portion). Representative curves from one assay run with three technical replicates are shown. RU, response units. (**c**) Binding of Dl1.72 to cell surface DLL1. CHO-k1 cells overexpressing rhDLL1 on the cellular surface (CHO-k1/DLL1^+^, filled circles) and CHO-k1 control cells not expressing hDLL1 (open circles) were incubated with Dl1.72 or Ctr Ab, and Ab-antigen binding was determined by flow cytometry. MFI, mean fluorescence intensity. Graphs in (**a**,**c**) are representative of three independent assays. (**d**) Binding of Dl1.72 to human ER^+^ BC tumor samples by IHC. Matched-tissue sections from ER^+^ BC samples (*n* = 9) were stained individually with Dl1.72 or Ctr Ab, pre-conjugated with HRP, or with the commercial anti-DLL1 polyclonal ab84620 followed by anti-rabbit HRP-conjugated secondary antibody as a control for DLL1 detection and tissue distribution. Representative IHC images from matched samples at disease stage T1 are shown. Brown color indicates DLL1 staining and blue nuclei counterstaining with hematoxylin. (**e**) Quantification of the mean value (± SEM) of each immunoreactive score. Scale bars, 50 µm. *, *p* < 0.05 (Mann–Whitney test).

**Figure 2 cancers-13-04074-f002:**
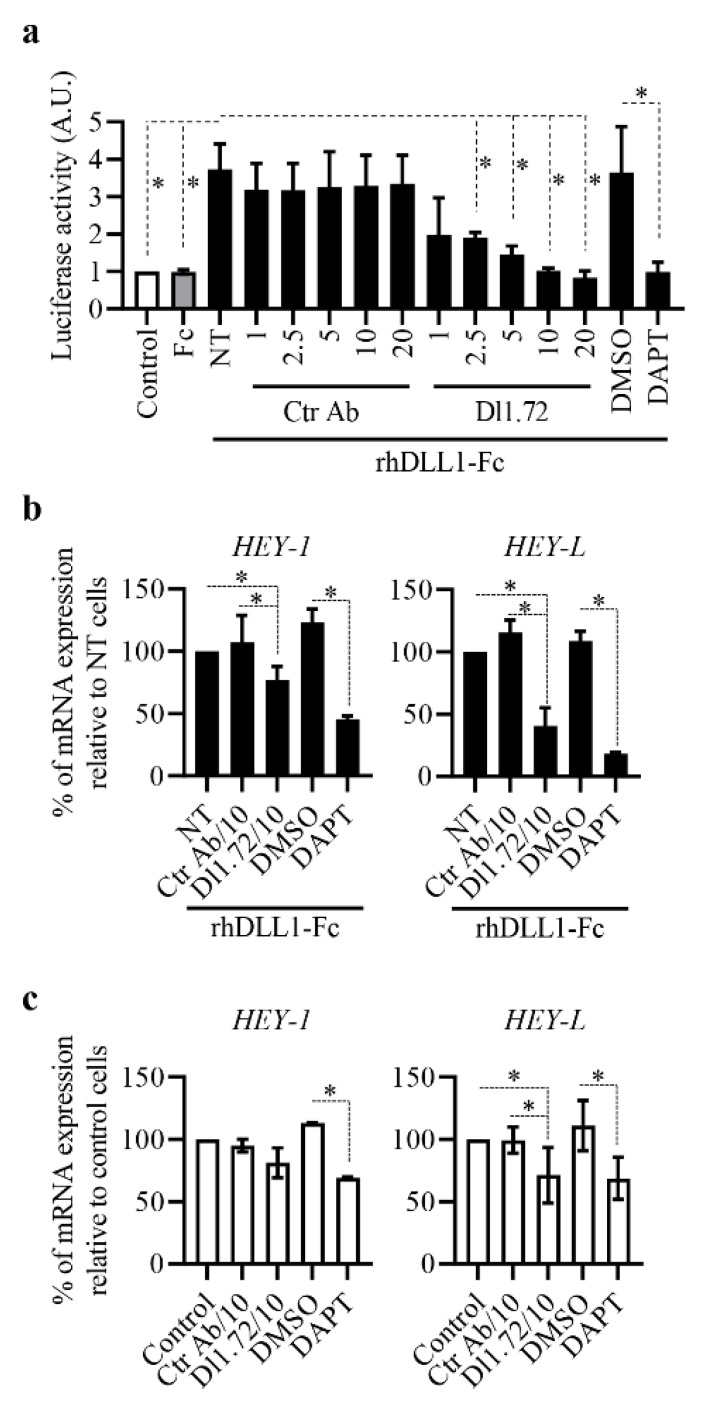
Anti-DLL1 Dl1.72 impairs DLL1-Notch signaling in MCF-7 cells. (**a**) Notch luciferase reporter assay. Cells transfected with the Notch firefly luciferase reporter were cultured for 28 h in uncoated (control) wells or wells pre-coated with Fc control protein or rhDLL1-Fc in the absence (NT) or presence of the indicated concentration (µg/mL) of Dl1.72, Ctr Ab, the pan-Notch inhibitor DAPT (5 µM) or DMSO (DAPT vehicle). The graph shows luciferase activity (mean ± SD, *n* = 3) compared to control unchallenged cells. (**b**) Effect of Dl1.72 on the levels of *HEY-1* and *HEY-L* induced by DLL1. Cells were cultured in wells pre-coated with rhDLL1-Fc proteins and not treated (NT) or treated with Dl1.72, Ctr Ab (10 µg/mL each), DAPT or DMSO. After 17–20 h, the transcript levels of Notch-dependent genes *HEY-1* and *HEY-L* were determined by RT-qPCR. Graphs show the mean percentage (±SD, *n* = 3) of *HEY-1* and *HEY-L* mRNAs relative to untreated cells. (**c**) Effect of Dl1.72 on the basal levels of *HEY-1* and *HEY-L*. MCF-7 cells were cultured in the absence (control) or presence of Dl1.72, Ctr Ab (10 µg/mL each), DMSO or DAPT, and the transcript levels were determined as above. Graphs show the mean percentage (±SD, *n* > 3) of *HEY-1* and *HEY-L* mRNAs relative to unstimulated control cells. *, *p* < 0.05 (two-tailed paired Student *t*-test).

**Figure 3 cancers-13-04074-f003:**
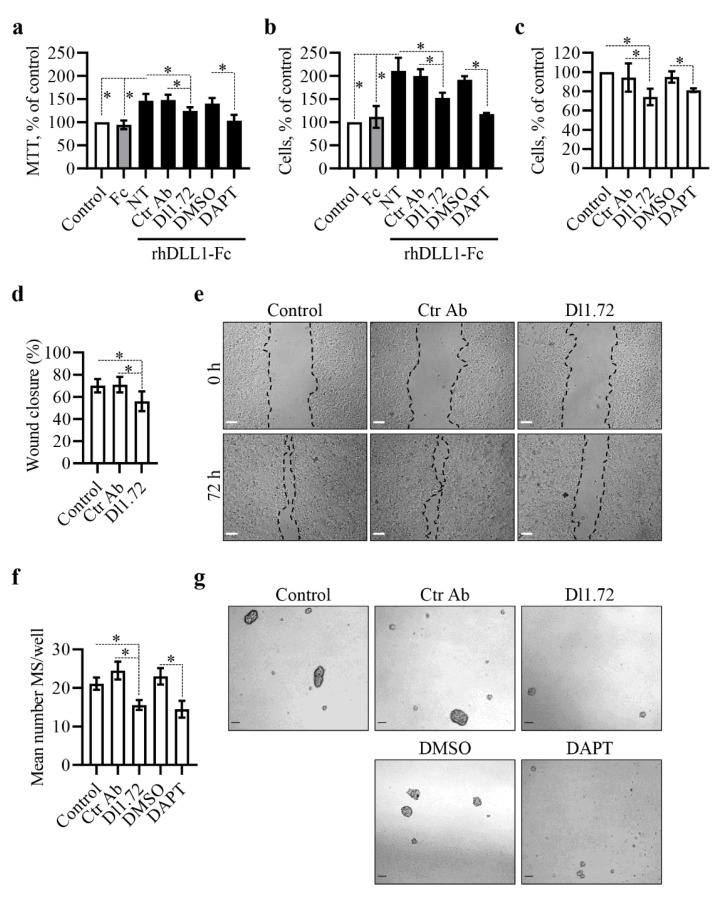
Anti-DLL1 Dl1.72 impairs MCF-7 cell proliferation, migration and mammosphere formation. (**a**,**b**) Effect of Dl1.72 in MCF-7 cell proliferation induced by DLL1. Cells were cultured in uncoated wells (control) or wells coated with Fc or rhDLL1-Fc proteins in the absence (NT) or presence of Dl1.72, Ctr Ab (10 µg/mL), DAPT or DMSO, and cell growth was analyzed after 72 h by the (**a**) MTT method and (**b**) trypan blue exclusion microscopy. Values were calculated as percentages relative to those obtained in control cells. The graphs represent mean percentage (±SD, *n* = 4). (**c**) Effect of Dl1.72 on cell proliferation in the absence of DLL1-induced proliferation. At 16 h following seeding, cells were left untreated or treated, as indicated above, and cell growth was analyzed after 72 h by trypan blue exclusion microscopy. The graph shows the mean percentage (±SD, *n* = 6) of cells relative to control untreated cells. (**d**,**e**) MCF-7 cells seeded and treated as in (**c**) were scratched at 90% confluence, and wound closure was evaluated by microscopy. (**d**) Mean percentage values (±SD, *n* = 3) of wound closure at 72 h after wounding. (**e**) Representative images taken at the indicated times post-wounding from the three independent experiments. Scale bars, 200 μm. (**f**,**g**) Effect of Dl1.72 on MCF-7 BCSC population. Single cell suspensions were plated in mammosphere (MS) formation medium in ultra-low attachment plates and not treated (control) or treated as in (**c**). (**f**) Graph shows mean number (±SD, *n* = 5) of MS after 5 days of culture. (**g**) Representative images from experiments. Scale bars, 50 μm. *, *p* < 0.05 (two-tailed paired Student *t*-test).

**Figure 4 cancers-13-04074-f004:**
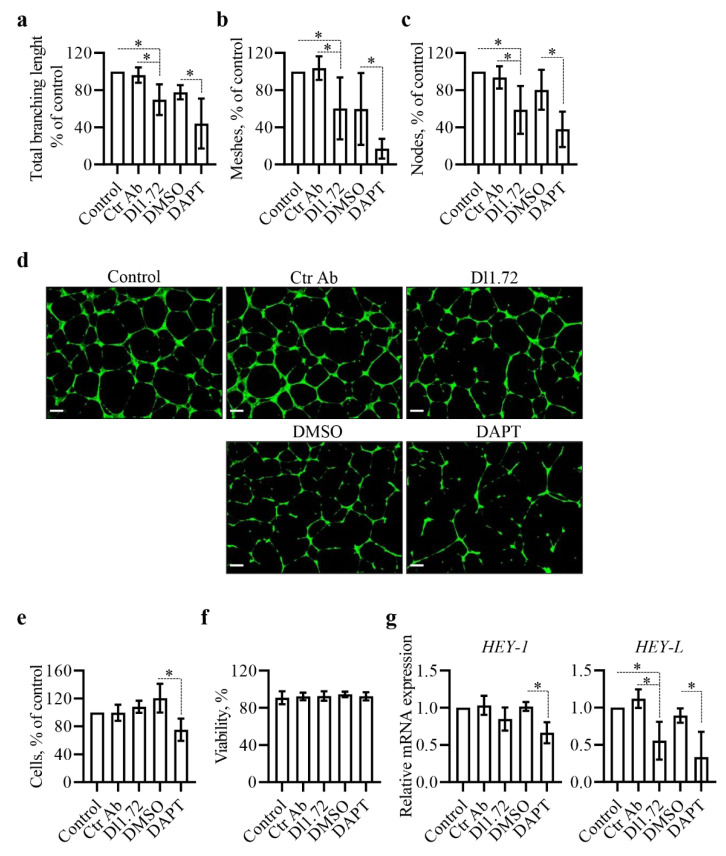
Anti-DLL1 Dl1.72 impairs the angiogenic potential of HUVEC cells. HUVEC cells were cultured in the absence (control) or presence of Dl1.72, Ctr Ab (10 µg/mL each), DAPT or DMSO for 70–76 h until 80% confluence, collected and plated on matrigel or geltrex to evaluate (**a**–**d**) HUVEC tube-network forming ability performance. The graphs in (**a**–**c**) show mean percentage values (±SD, *n* = 6) of total branching length, meshes/cavities, and nodes. (**d**) Representative images of calcein-stained cells. Scale bars, 200 μm. Mean percentage of (**e**) cells and (**f**) viability relative to control untreated cells (±SD) from five independent assays performed as described above. (**g**) Gene expression profile of Notch-target genes *HEY-1* and *HEY-L* in HUVEC cells treated as above was determined by RT-qPCR. Graphs show mean mRNA fold change relative to control cells (±SD, *n* = 4). *, *p* < 0.05 (two-tailed paired Student *t*-test).

**Figure 5 cancers-13-04074-f005:**
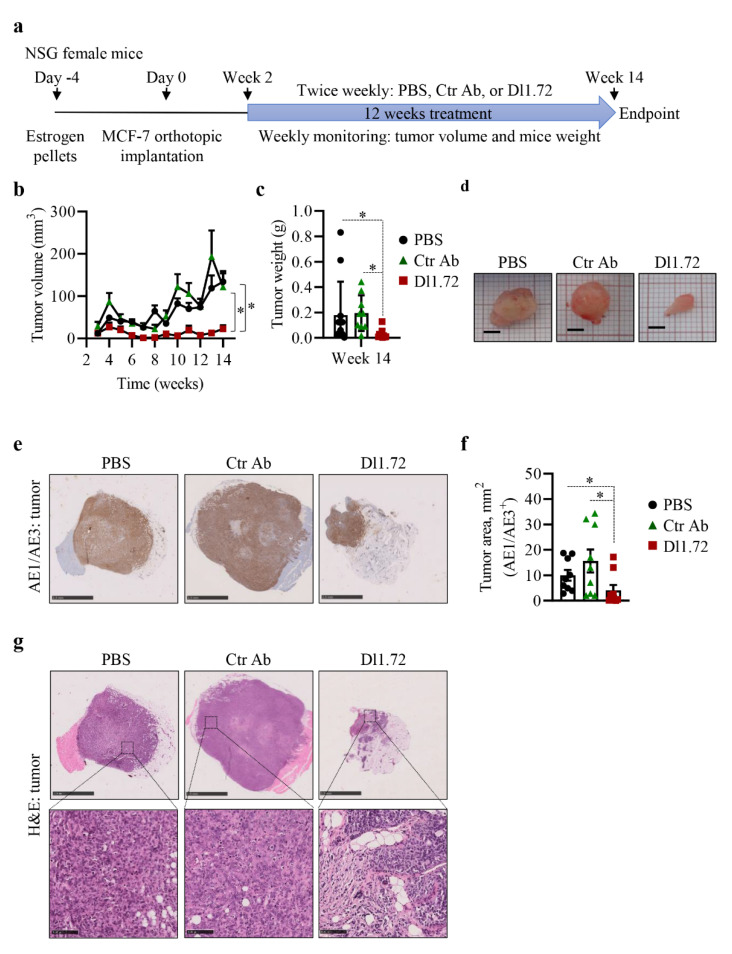
Anti-DLL1 Dl1.72 decreases tumor growth in a MCF-7 xenograft mice model. (**a**) Schematic representation of mice tumor induction and treatment with either PBS (control vehicle) or with 10 mg per kg of isotype-matched Ctr Ab or Dl1.72 for 12 weeks. (**b**) Tumor growth was evaluated by caliper (PBS, *n* = 12; Ctr Ab, *n* = 9; Dl1.72, *n* = 10). (**c**) Weight of tumors from each group immediately after resection. (**d**) Representative whole tumor images from each group. (**e**) Representative IHC images of the largest cross-sections of tumors from each group stained with the pan-cytokeratin AE1/AE3 Ab, specific to human epithelial cells. Scale bars, 2.5 mm. (**f**) Quantification of tumor area determined by AE1/AE3 staining (*n* = 9 tumors/group). (**g**) Representative hematoxylin and eosin (H&E) stained images of tumor cross-sections of each group. Scale bars, 2.5 mm (top image) and 100 µm (bottom image). Data are expressed as mean values ± SEM. *, *p* < 0.05 by one-way ANOVA with Tukey’s multiple comparisons test in (**a**) and the Mann–Whitney test in (**c**,**f**).

**Figure 6 cancers-13-04074-f006:**
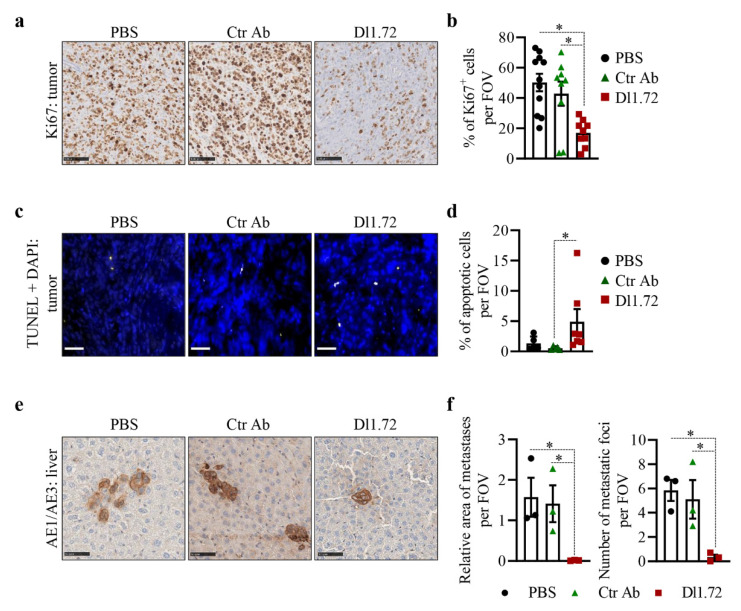
Treatment with anti-DLL1 Dl1.72 leads to reduced tumor cell proliferation, increased tumor cell apoptosis and impaired liver micrometastases in MCF-7 xenografts. (**a**) Representative IHC images of cross-sectioned tumors from mice treated with either PBS, Ctr Ab or Dl1.72 twice weekly for 12 weeks stained with Ki67 (proliferation marker). Scale bars, 100 µm. (**b**) Quantification of Ki67 positive cells from the stained sections (PBS, *n* = 11 tumors; Ctr Ab and Dl1.72, *n* = 9 tumors/each group). (**c**) Representative IF images of cross-sectioned tumors from each group stained with TUNEL method (apoptotic cells, yellow) and DAPI (nuclei, blue). Scale bars, 50 µm. (**d**) Quantification of apoptotic cells (PBS and Ctr Ab, *n* = 5 tumors/each group; Dl1.72, *n* = 7 tumors). (**e**) IHC representative images of cross-sectioned livers stained with pan-cytokeratin AE1/AE3 antibody. Scale bars, 100 µm. (**f**) Quantification of liver relative areas (mm^2^) occupied by AE1/AE3^+^ cells and number of liver micrometastatic foci (i.e., number of AE1/AE3^+^ areas) of indicated groups (*n* = 3 tumors/group). All quantifications were performed in ten random fields of view (FOVs) at 10x magnified images from each tumor (**b**,**d**) and liver (**f**) cross-section. Data are expressed as mean values ± SEM. *, *p* < 0.05 (Mann–Whitney test).

**Table 1 cancers-13-04074-t001:** Estimated EC_50_ values and affinity constants of anti-DLL1 Dl1-72 mAb towards DLL1. Data are mean value ± SD of three independent ELISA and flow cytometry assays and three replicates from the SPR assays.

ELISA	Flow Cytometry	SPR		
EC_50_ (nM)	EC_50_ (nM)	K_D_ (nM)	*k*_a_ (s^−1^)	*k*_d_ (M^−1^ s^−1^)
1.15 ± 0.16	2.07 × 10^−2^ ±1.20 × 10^−2^	4.78 ± 1.05	5.44 × 10^5^ ±8.21 × 10^4^	2.60 × 10^−3^ ±2.50 × 10^−4^

## Data Availability

The data presented in this study are available in the article or Appendix A.

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
