# Peer review of "Development of Dl1.72, a Novel Anti-DLL1 Antibody with Anti-Tumor Efficacy against Estrogen Receptor-Positive Breast Cancer"

_cancers, 2021, doi:10.3390/cancers13164074_

Round 1

Reviewer 1 Report

Overexpression of DLL1 is associated with a poor prognosis of BC. The authors proposed an anti-DLL1 antibody fragment that was converted into a full human IgG1 (Dl1.72) against ER+ BC to handle the prognosis of the disease. The study characterized the in vitro and in vivo BC anti-tumorigenic effect of a novel anti-DLL1 antibody. I think that the study is interesting and will gain the attention of the research community. the results support the hypothesis.

Author Response

Reviewer 1

Overexpression of DLL1 is associated with a poor prognosis of BC. The authors proposed an anti-DLL1 antibody fragment that was converted into a full human IgG1 (Dl1.72) against ER+ BC to handle the prognosis of the disease. The study characterized the in vitro and in vivo BC anti-tumorigenic effect of a novel anti-DLL1 antibody. I think that the study is interesting and will gain the attention of the research community. the results support the hypothesis.

Author’s reply to the reviewer:

We would like to thank the reviewer for her/his positive comments on the manuscript.

Reviewer 2 Report

This original research article entitled “Development of Dl1.72, a novel anti-DLL1 antibody with anti-tumor efficacy against estrogen receptor-positive breast cancer” by Silva et al., developed a monoclonal antibody Dl1.72.

Next, authors conducted series of experiments to show that the Dl1.72 antibody can effectively targets activated Notch signaling in breast cancer cells by specifically inhibiting DLL1.

Finally, authors showed that Dl1.72 significantly inhibited tumor growth and liver metastases in a xenograft mouse model, without apparent toxicity, is a major strength to their study.

Overall this is a well-designed study and the experiments were performed elegantly.

However, refer my comment below for editing the manuscript.

Comments:

  1. Methods were too long. Cut it short and consider providing them in supplemental
  2. Remove technical details from the results section. Example remove line 408 -411 in page #11.
  3. Figure 1d – remove technical replicates T2 and T4. Consider providing them in supplemental.
  4. Figure 1C, what is AU?

Author Response

Reviewer 2

This original research article entitled “Development of Dl1.72, a novel anti-DLL1 antibody with anti-tumor efficacy against estrogen receptor-positive breast cancer” by Silva et al., developed a monoclonal antibody Dl1.72.

Next, authors conducted series of experiments to show that the Dl1.72 antibody can effectively targets activated Notch signaling in breast cancer cells by specifically inhibiting DLL1.

Finally, authors showed that Dl1.72 significantly inhibited tumor growth and liver metastases in a xenograft mouse model, without apparent toxicity, is a major strength to their study.

Overall this is a well-designed study and the experiments were performed elegantly.

However, refer my comment below for editing the manuscript.

Comments:

  1. Methods were too long. Cut it short and consider providing them in supplemental
  2. Remove technical details from the results section. Example remove line 408 -411 in page #11.
  3. Figure 1d – remove technical replicates T2 and T4. Consider providing them in supplemental.
  4. Figure 1C, what is AU?

Author’s reply to the reviewer:

We would like to thank the reviewer for her/his positive comments on the manuscript. We agree with the comments of the reviewer, and we reply to them below:

  1.  Methods were too long. Cut it short and consider providing them in supplemental

Reply: We agree with the reviewer and we thank for his/her observation. We have removed methodological details relative to:

  1. Phage Display technology (removed lines:129-140);
  2. Real Time PCR (removed lines: 213-215);
  3. Mammosphere formation assay: we removed that methodological part and referred to the methodology already reported in our previous manuscript (reference 22, deleted lines: 234-239).

In addition, some of the detailed text explaining some of the methods used were removed from the main text and placed them as Supplementary Material, as suggested by the reviewer. In particular, we shifted the following methodology: Surface Plasmon Resonance (lines 157-171); Horseradish Peroxidase Anti-DLL1 Dl1.72 Conjugation (removed lines: 172-181; now shortened in the revised version, lines 153-156, details in supplementary methodology); SDS-PAGE and Western blot analysis (removed lines 182-189); Immunohistochemical Analysis (removed lines, 296-313 and 326-337). The new supplementary information file has been provided along with the revised manuscript.

  1. Remove technical details from the results section. Example remove line 408 -411 in page #11.

We agree with the reviewer. We revised our result section and remove excessive details which, after reading the reviewer comment, we realized they were reducing the overall flow of the manuscript:

We removed line 408-411, as suggested by the reviewer, as well as:

Line 378;

Lines 387-389;

Lines 450-453;

Lines 597-599.

  1. Figure 1d – remove technical replicates T2 and T4. Consider providing them in supplemental.

We have removed the technical replicates, showing only one and placing the other in Supplementary Material (now indicated in the line 422);

  1. Figure 1C, what is AU?

In fact, AU in Figure 1C was included by mistake. MFI is the correct one. We apologize the reviewer for this. This has been corrected in the present version of the Figure 1C.
